# Structural basis of neuropeptide Y signaling through Y1 receptor

Chaehee Park [1,7], Jinuk Kim[1,7], Seung-Bum Ko[1], Yeol Kyo Choi [2], Hyeongseop Jeong [3], Hyeonuk Woo[4], Hyunook Kang [1], Injin Bang [1,6], Sang Ah Kim[1,5], Tae-Young Yoon [1,5], Chaok Seok [4], Wonpil Im [2] & Hee-Jung Choi [1✉]

Neuropeptide Y (NPY) is highly abundant in the brain and involved in various physiological processes related to food intake and anxiety, as well as human diseases such as obesity and cancer. However, the molecular details of the interactions between NPY and its receptors are poorly understood. Here, we report a cryo-electron microscopy structure of the NPY-bound neuropeptide Y1 receptor ($Y_1R$) in complex with $G_{i1}$ protein. The NPY C-terminal segment forming the extended conformation binds deep into the $Y_1R$ transmembrane core, where the amidated C-terminal residue Y36 of NPY is located at the base of the ligand-binding pocket. Furthermore, the helical region and two N-terminal residues of NPY interact with $Y_1R$ extracellular loops, contributing to the high affinity of NPY for $Y_1R$. The structural analysis of NPY-bound $Y_1R$ and mutagenesis studies provide molecular insights into the activation mechanism of $Y_1R$ upon NPY binding.

[1] Department of Biological Sciences, Seoul National University, Seoul 08826, Republic of Korea. [2] Departments of Biological Sciences and Chemistry, Lehigh University, Bethlehem, PA 18015, USA. [3] Center for Electron Microscopy Research, Korea Basic Science Institute, Chungcheongbuk-do 28119, Republic of Korea. [4] Department of Chemistry, Seoul National University, Seoul 08826, Republic of Korea. [5] Institute for Molecular Biology and Genetics, Seoul National University, Seoul 08826, Republic of Korea. [6] Present address: Perlmutter Cancer Center, New York University Langone Health, and Department of Biochemistry and Molecular Pharmacology, New York University School of Medicine, New York 10016 NY, USA. [7] These authors contributed equally: Chaehee Park, Jinuk Kim. ✉email: choihj@snu.ac.kr

The human neuropeptide Y (NPY) system comprises three peptide ligands, NPY, peptide YY (PYY), and pancreatic polypeptide (PP), and four functional NPY receptors $Y_1R$, $Y_2R$, $Y_4R$, and $Y_5R$[1]. These endogenous peptide ligands consisting of 36 amino acids with the amidated C-terminus activate specific NPY receptors generally coupled to $G_i$ or $G_o$ protein[2]. Of these three peptide ligands, NPY, a highly abundant peptide ligand in the brain, can activate all four subtypes of the NPY receptor and is involved in various physiological processes such as food intake, stress response, anxiety, and memory retention[3–5]. Furthermore, NPY signaling is involved in human diseases such as obesity, mood disorders, and cancers[6–8].

The nuclear magnetic resonance (NMR) structure of NPY reveals that its C-terminal segment (13–36) forms an amphipathic α-helix, and the remaining N-terminal part is unstructured and flexible[9,10]. Previous functional assays using the N-terminal truncation mutants of NPY indicate that the complete N-terminus of NPY is necessary for $G_i$ signaling through $Y_1R$ but not through $Y_2R$, which shares approximately 30% sequence identity with $Y_1R$[10,11]. Similarly, PYY, highly homologous to NPY, binds to $Y_1R$ and $Y_2R$ in its full-length form; however, PYY (3–36), the N-terminal cleaved form found in the circulation, binds only to $Y_2R$[12–14].

NPY receptors belong to the β subgroup of class A G protein-coupled receptors (GPCRs). Among them, $Y_1R$ is expressed in the central nervous system (CNS) as well as in the adipose tissue and vascular smooth muscle cells, where it leads to enhanced cell proliferation and the induction of food intake upon activation by NPY[15–17]. Thus, the $Y_1R$ antagonist has been proposed as a potential drug for treating obesity[16]. Furthermore, $Y_1R$ is highly expressed in human primary breast cancer, implying the utility of $Y_1R$ as a diagnostic marker for breast cancer[18].

The crystal structures of the small molecule antagonist-bound $Y_1R$ have been published recently[10]. Although these structures provide molecular details of $Y_1R$-selective antagonist binding and the overall architecture of an inactive state of $Y_1R$, the molecular mechanism of its activation by binding of the endogenous agonist NPY is still unknown.

Here, we present a single-particle cryo-electron microscopy (cryo-EM) structure of NPY-bound wild-type $Y_1R$ with $G_{i1}$ protein coupled. The structure reveals that five amino acids at the C-terminus of NPY form an extended conformation and are inserted into a pocket formed by the transmembrane (TM) domain of $Y_1R$. In addition, the helical region and N-terminus of NPY are shown to be involved in $Y_1R$ binding. Together with molecular dynamics (MD) simulations and functional analysis of various mutations, our structure provides molecular details of endogenous peptide recognition by $Y_1R$ and suggests the activation mechanism of $Y_1R$ upon NPY binding.

## Results

**Structure determination of the NPY–$Y_1R$–$G_{i1}$ complex.** The published antagonist-bound crystal structure of $Y_1R$ was solved using a modified construct involving thermostabilizing mutation, C-terminal truncation, and replacement of intracellular loop 3 (ICL3) with T4 lysozyme[10]. In our study, we used the wild-type $Y_1R$ construct (2–384) with minimal engineering (such as affinity tag) to solve the receptor structure, thereby facilitating molecular analysis of the activation mechanism of the native receptor. $Y_1R$ is known to couple $G_{i1}$ protein to activate downstream signaling upon NPY binding[2]. Before purifying the NPY–$Y_1R$–$G_{i1}$ complex, we confirmed the functionality of the synthesized NPY by performing bioluminescence resonance energy transfer (BRET) assays, which showed that $Y_1R$ specifically recruited $G_{i1}$ in response to NPY binding (Supplementary Fig. 1). For the

structural study, $Y_1R$ and $G_{i1}$ heterotrimer were purified separately and mixed in the presence of NPY. The complex was incubated with apyrase to obtain a nucleotide-free $G_{i1}$ heterotrimer, and single-chain variable fragment termed scFv16, that specifically recognizes heterotrimeric $G_i$ was added as a stabilizer[19] (Supplementary Fig. 2).

We determined a cryo-EM structure of the NPY–$Y_1R$–$G_{i1}$–scFv16 complex at a nominal resolution of 3.2 Å in glyco-diosgenin (GDN) micelles (Supplementary Table 1 and Supplementary Fig. 3). Similar to other structures of nucleotide-free G protein heterotrimers bound to GPCR, the α-helical domain of $G_{i1}$ was not modeled in our structure because of its flexibility. Inspection of the cryo-EM map clearly showed the density of NPY protruding into the extracellular region (Fig. 1a). To further focus on the binding of NPY to $Y_1R$, we subtracted the heterotrimeric G protein signal and subjected to local refinement (Supplementary Fig. 3) as used in the cryo-EM analysis of secretin-bound secretin receptor–$G_s$ complex[20]. The resulting cryo-EM map allowed modeling of the five residues at the N-terminus (1–5) and the C-terminal half of NPY (20–36) (Fig. 1b, c and Supplementary Fig. 4).

**Overall structure of NPY-bound $Y_1R$ receptor.** Structural comparison of the NPY-bound and antagonist-bound $Y_1R$ reveals distinct conformational changes in the extracellular region, TM core, and cytoplasmic part of the receptor upon activation. As the amidated C-terminus of NPY penetrates deep into the TM core, the extracellular tips of TM3, TM4, TM6, and TM7 slightly move outward by 1.9–2.5 Å in the NPY-bound $Y_1R$ structure compared to the antagonist-bound structure, opening up the ligand-binding pocket (Fig. 2a). In fact, the calculated solvent-accessible ligand-binding cavities in the antagonist-bound and NPY-bound structures are ~506 and 730 Å$^3$, respectively (Supplementary Fig. 5). Besides the interaction of the C-terminal tail of NPY with the $Y_1R$ TM core, the molecular interaction of the remaining NPY with $Y_1R$ was difficult to characterize, as the NPY map was not well resolved (Supplementary Fig. 3). However, we could observe the density of the N-terminal and helical regions of NPY, which are surrounded by extracellular loops (ECLs) and the N-terminal region of $Y_1R$ (Supplementary Fig. 6). ECL2, ECL3, and the N-terminal region of $Y_1R$ directly interact with NPY to varying degrees. The details of the NPY interaction are discussed later.

NPY binding results in the TM core of $Y_1R$ undergoing conventional conformational changes associated with the activation mechanism of class A GPCRs. Immediately below the C-terminus of NPY peptide, side chains of I128$^{3.40}$, P223$^{5.50}$, and F272$^{6.44}$ (superscripts are the Ballesteros–Weinstein numbers[21]) in the connector region are repacked to contract the interface of TM3, 5, and 6 (Fig. 2b), resulting in the outward movement of TM6 and inward movement of TM7 at the cytoplasmic part of $Y_1R$ (Fig. 2a). In addition, R138$^{3.50}$ of the (D/E)R(Y/H) motif makes close contact with Y231$^{5.58}$ and Y320$^{7.53}$ of the NPxxY motif (Fig. 2c), a well-known key interaction observed in the activated GPCR structures. These conformational changes demonstrate that our structure represents the active conformation of class A GPCRs.

Compared to other $G_{i1}$-bound class A GPCRs, such as neurotensin receptor 1 (NTSR1) and μ-opioid receptor (μOR), $Y_1R$ has a relatively short ICL2, comprising six amino acids, including two Pro residues (Fig. 2d). In NTSR1 and μOR, ICL2s adopt an α-helix upon activation, interacting with the αN, αN-β1 loop and α5 helix of $G_i$[22,23]. In contrast, ICL2 of $Y_1R$ remains as a loop in the $G_{i1}$-bound state, forming contacts with only the α5 helix of $G_i$ through R146$^{ICL2}$ and R149$^{ICL2}$ (Fig. 2d). In addition to ICL2, hydrophobic residues in TM3, TM5, and TM6 (I142$^{3.54}$, I234$^{5.61}$, L238$^{5.65}$, I261$^{6.33}$, and L265$^{6.37}$) of $Y_1R$ form van der

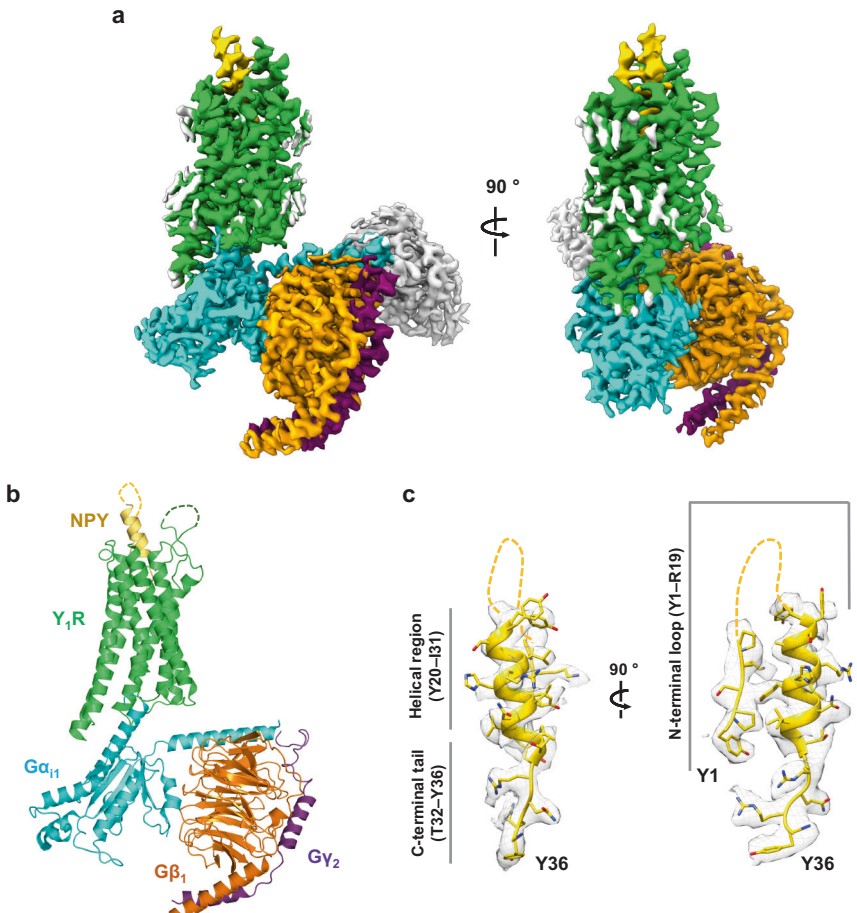

**Fig. 1 Overall structure of the NPY–Y₁R–Gᵢ₁–scFv16 complex. a** The cryo-EM map of the NPY–Y₁R–Gᵢ₁–scFv16 complex is shown. Y₁R, NPY, Gα_{i1}, Gβ_{1}, Gγ_{2}, and scFv16 are colored green, yellow, cyan, orange, purple, and gray, respectively. The remaining micelle density is shown in light gray. Details on cryo-EM map generation are described in the "Methods" section. **b** The structure of the NPY–Y₁R–Gᵢ₁ complex is shown. scFv16 is included in the final structure but omitted in this figure for clarity. **c** The sharpened cryo-EM map with the NPY peptide model is shown in two orientations. NPY residues 6–19 are not modeled in this structure and are shown as dashed lines. The three regions of NPY are marked as the N-terminal loop region (Y1–R19), the helical region (Y20–I31), and the C-terminal tail (T32–Y36).

Waals interactions with L348 and L353 in the α5 helix of Gᵢ₁ (Supplementary Fig. 7) and R260^{6.32} of Y₁R forms polar interaction with the carboxyl group of F354 in the α5 helix of Gᵢ₁ (Supplementary Fig. 7). Most of these interactions are conserved in the hNTSR1-Gᵢ₁ and μOR-Gᵢ₁ structures (Supplementary Fig. 7). However, when aligning the receptors, the relative position of the α5 helix of Gᵢ₁ bound to Y₁R slightly differs by ~2 Å displacement of the C-terminus of Gα_{i1} or ~8° tilt angle of the α5 helix of Gᵢ₁ in the NTSR1-bound and μOR-bound Gᵢ₁ structures, respectively (Supplementary Fig. 7).

**Binding of the C-terminal tail of NPY to Y₁R.** Our cryo-EM map shows a well-resolved density for the five C-terminal residues of NPY (32–36) (Fig. 1c), forming an extended structure, contrasting the α-helix formation of residues 13–36 in the NMR structure of human NPY (PDB ID 1RON)[9]. However, helix unwinding at the C-terminal tail of NPY was not unexpected since the previous NMR study of porcine NPY bound to Y₁R suggested an extended conformation of the NPY C-tail[10]. Similarly, the nine amino acids at the C-terminus of orexin-B neuropeptide (OxB) were previously shown to form an extended conformation in the orexin receptor (OX₂R)-bound state but form an α-helix in the receptor-free state[24,25].

The extended conformation of the NPY C-tail binds to a pocket lined by TM helices 2–7, with a depth of ~11 Å from the top surface of the membrane (Fig. 3a, b). It is well established that the amidation of NPY C-terminal tyrosine is critical for its function; consistently, in this study, non-amidated NPY failed to elicit G protein signaling (Supplementary Fig. 8). At the bottom of the ligand-binding pocket, the C-terminal amide of NPY points toward the side chain of Q120^{3.32} (Fig. 3c), which has been predicted to interact with the Y36 side chain in the NPY-docked Y₁R model[10]. The importance of Q120^{3.32} for Gᵢ recruitment and signaling upon NPY treatment was investigated using BRET and calcium signaling assays. As expected, the Q120^{3.32}A mutant exhibited reduced Gα_{i1} recruitment and an increased EC_{50} (Supplementary Table 2 and Supplementary Figs. 9, 10, and 11). The C-terminal amide of Y36 is further coordinated by H306^{7.39} through polar interactions and C93^{2.57} and M310^{7.43} through van der Waals contacts (Fig. 3c). Furthermore, the ligand-binding pocket of Y₁R is highly acidic (Supplementary Fig. 12). The acidic residues of the binding pocket repel the negatively charged C-terminus and favor the amidated C-terminus. A similar acidic patch in the ligand-binding pocket is observed in OX₂R, where the amidated C-terminus of OxB binds. In contrast, the ligand-binding pocket is basic in NTSR1 and endothelin B receptor (ET_{B}R), whose peptide agonists have the C-terminal carboxyl groups (Supplementary Fig. 12).

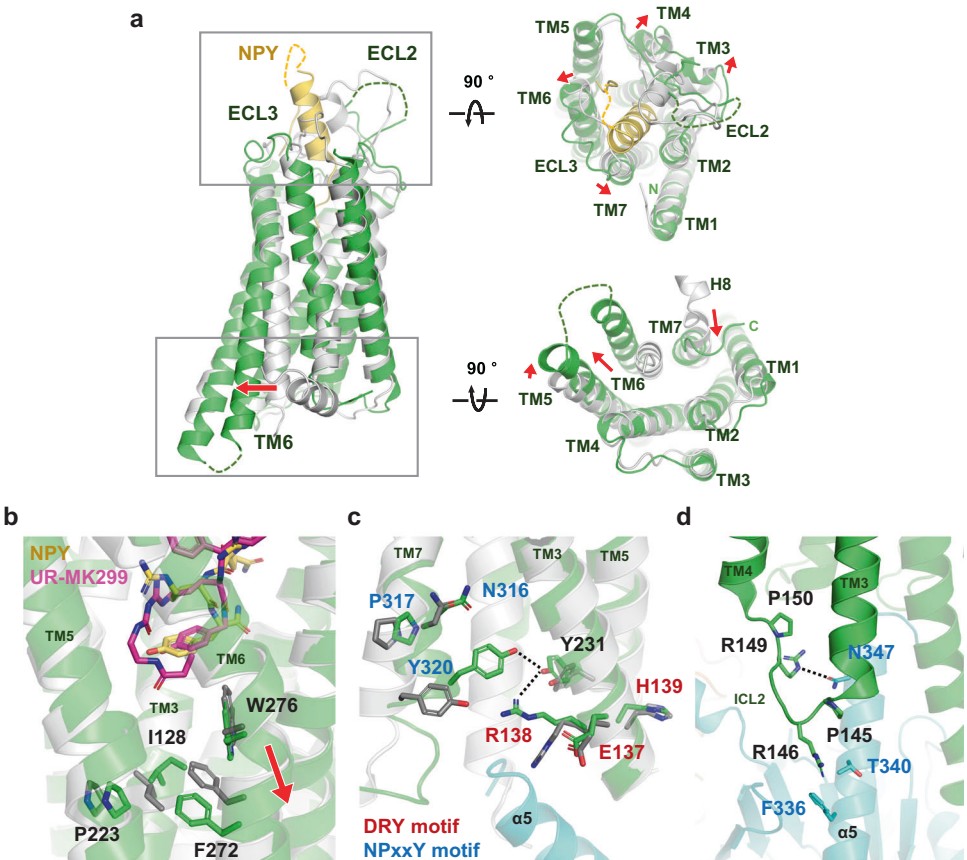

**Fig. 2 Comparison between NPY-bound and antagonist-bound Y₁R structures. a** Structural alignment between NPY (yellow)-bound Y₁R (current structure, green) and antagonist-bound Y₁R (PDB ID 5ZBQ, light gray, residues 18–32 of Y₁R and antagonist are omitted for clarity) clearly shows characteristic TM6 movement as observed in the active structures of class A GPCRs. Views from extracellular (right-upper panel) and cytoplasmic (right-lower panel) sides are shown on the right. Each red arrow represents the movement of the TM helix. **b** Structural changes at the connector region (P$^{5.50}$I$^{3.40}$F$^{6.44}$ motif) upon NPY binding are shown. The antagonist-bound Y₁R (PDB ID 5ZBQ) is colored in light gray and an antagonist, UR-MK299, in magenta. **c** Structural changes at the D(E)/R/Y(H) (labeled in red) and NPxxY (labeled in blue) motifs are shown. **d** The binding interface between α5 of G$_{i1}$ (in cyan) and ICL2 is shown. Residues participating in the interactions and two Pro residues (P145$^{ICL2}$ and P150$^{ICL2}$) in ICL2 are shown as sticks. Dashed lines represent the polar interactions.

The Y36 side chain of NPY forms a hydrogen bond with Q219$^{5.46}$ through its hydroxyl group and hydrophobic interaction with I124$^{3.36}$ through its phenyl ring (Fig. 3c). The importance of Q219$^{5.46}$ and I124$^{3.36}$ for NPY signaling was demonstrated by 13.5-fold and 3.5-fold reduction in NPY potency in the Q219$^{5.46}$A and I124$^{3.36}$A mutants, respectively[10]. Previous mutational studies of NPY showed that the Y36F mutation had a relatively mild effect on Y₁R binding[26]. In contrast, the Y36A mutation caused a loss of binding to Y₁R[10,26], suggesting that the phenyl ring of Y36 is more important for NPY signaling. Our structure shows that the phenyl ring of Y36 forms an intramolecular interaction with R35 of NPY (Supplementary Fig. 13), which aids in correctly positioning R35, in addition to hydrophobic interactions with Y₁R (Fig. 3a).

R33 and R35 of NPY extensively interact with the Y₁R TM core, and alanine scanning mutagenesis of NPY has shown that R33A and R35A mutations of NPY exhibit the most severe defects in Y₁R binding[10,26]. R33 of NPY forms a hydrogen bond with N283$^{6.55}$ and π–cation interactions with F286$^{6.58}$ and F302$^{7.35}$ (Fig. 3d). BRET analysis using the N283$^{6.55}$A mutant showed a dramatic decrease in Gα$_i$ recruitment and reduced NPY potency by 85-fold compared to wild-type Y₁R (Supplementary Table 2 and Supplementary Figs. 9, 10, and 11). Previous mutagenesis studies of F286$^{6.58}$ and F302$^{7.35}$ showed that substituting of these residues with alanine caused a reduction in

NPY potency[10,27]. R35 forms electrostatic interaction with D287$^{6.59}$ and van der Waals interaction with F173$^{4.60}$, both of which have been reported to be essential for NPY signaling[10,28].

Q34 and T32 of NPY are oriented opposite to the R35 and R33 side chains, interacting with T97$^{2.61}$, Y100$^{2.64}$, and the backbone carbonyl of D104$^{2.68}$ (Fig. 3e). In particular, Y100$^{2.64}$ is sandwiched between T32 and Q34, forming nonpolar contacts with both (Fig. 3e). Consistent with these structural data, Y100$^{2.64}$A, a well-known mutation, dramatically decreased downstream NPY signaling[10,27,28]. In the antagonist-bound structures, none of these residues that interact with T32 and Q34 are involved in antagonist binding.

Altogether, our structural analysis shows that the C-terminal tail of NPY forms extensive interactions with Y₁R residues on TM2, 3, 5, and 6. Previous and current mutational studies support the importance of these interacting residues in NPY signaling.

**Structural changes in Y₁R TM core by binding of NPY C-terminal tail.** The binding pocket for the NPY C-terminus is where the two antagonists (UR-MK299 and BMS-193885) bind in the previously reported inactive Y₁R structures. The structural comparison reveals that the hydroxyphenyl group and guanidine moiety of UR-MK299 exhibit binding modes similar to those of the side chains of Y36 and R35 of NPY, respectively, making polar interactions with Q219$^{5.46}$ and D287$^{6.59}$ (Fig. 4a). The BMS-

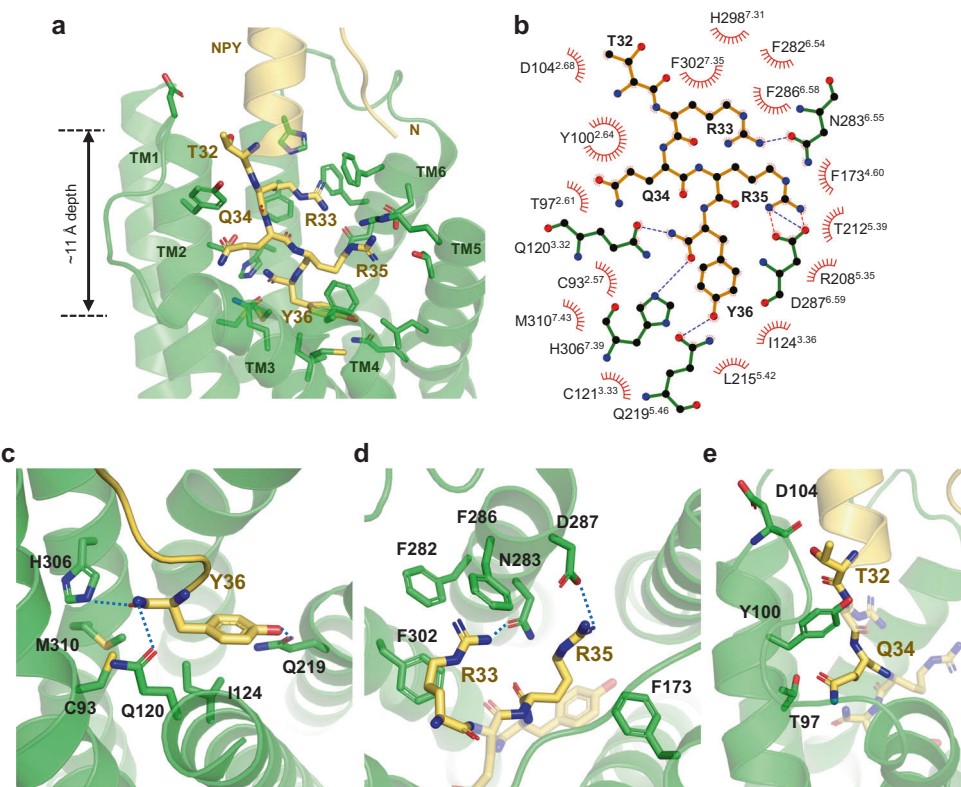

**Fig. 3 Binding of NPY C-terminal tail to Y₁R. a** The C-terminal tail of NPY (yellow) and Y₁R residues that participate in NPY binding are presented as sticks. **b** Schematic representation of NPY–Y₁R interactions analyzed using the LigPlot⁺ program is shown. The hydrogen bonds and the salt bridges are shown in blue and red dashed lines, respectively. **c**–**e** Detailed interactions between C-terminal residues of NPY and Y₁R are shown. The polar contacts are shown as blue dashed lines.

193885-bound inactive Y₁R structure shows that D287⁶·⁵⁹ of Y₁R interacts with the antagonist, whereas Q219⁵·⁴⁶ of Y₁R is not involved in this antagonist binding, as Q219⁵·⁴⁶ is pushed away by the dihydropyridine group (Fig. 4b). In both inactive structures, Q120³·³² makes van der Waals contact with M310⁷·⁴³, although Q120³·³² rotamers differ in two inactive structures, forming polar interaction with BMS-193885 or van der Waals interaction with W276⁶·⁴⁸ (Fig. 4c). In the NPY-bound Y₁R structure, the Q120³·³² sidechain adopts an upward-facing rotamer, forming a polar contact with the amidated C-terminus. Also, Q120³·³² no longer interacts with M310⁷·⁴³ (>7 Å) but with C93²·⁵⁷. Reorganization of interaction network near Q120³·³² by NPY binding stabilizes the conformation of the upward displacement of TM3 (Fig. 4c).

The hydrophobic moieties of the antagonists form hydrophobic networks with I124³·³⁶, I128³·⁴⁰, F272⁶·⁴⁴, W276⁶·⁴⁸, and L279⁶·⁵¹ at the bottom of the ligand-binding pocket and with F282⁶·⁵⁴, F286⁶·⁵⁸, and F302⁷·³⁵ near the entrance to the binding pocket, to stabilize the inactive Y₁R structure. These hydrophobic networks are rearranged in the NPY-bound structure. As mentioned above, at the bottom of the ligand-binding pocket, a rotamer change of I128³·⁴⁰ and repacking of F272⁶·⁴⁴ and W276⁶·⁴⁸ occur upon NPY binding (Fig. 2b). The three phenylalanine residues F282⁶·⁵⁴, F286⁶·⁵⁸, and F302⁷·³⁵ on TM6 and TM7 form a stable π–π network with phenyl groups present in both antagonists; however, this aromatic network is disrupted in the NPY-bound structure. The phenyl ring of F286⁶·⁵⁸ flips to form van der Waals contact with L30 and a π–cation interaction with R33 of NPY (Fig. 4d). F282⁶·⁵⁴ and F302⁷·³⁵ form a new interaction network with H298⁷·³¹ and I293ᴱᶜᴸ³ (Fig. 4e). F286⁶·⁵⁸ also interacts with Y1 and P2 of NPY. The NPY N-terminus interactions are discussed in the next section.

Comparison of binding modes between the NPY C-terminal tail and antagonists reveals common interactions of the ligand for

Y₁R binding and NPY-specific interactions, providing hints for designing novel Y₁R ligands that bind to the TM core. In addition, the structural comparison of the TM core between inactive and active states of Y₁R suggests the key events of conformational changes during activation by NPY, rearrangement of the hydrophobic network around F286⁶·⁵⁸ at the entrance to the ligand-binding pocket, and rearrangement of interaction network around Q120³·³² and I128³·⁴⁰ in the connector region at the bottom of the ligand-binding pocket.

**Binding of the N-terminal and helical regions of NPY to Y₁R.** In the OxB-bound OX₂R structure, the N-terminus of OxB was not observed despite being necessary for signaling. In our complex structure, we were surprised to observe the density of the helical and the N-terminal loop regions of NPY in addition to the C-terminal tail (Fig. 1c). The NPY residues 20–31 were built as an α-helix based on the continuous cryo-EM density from the C-terminus (Fig. 1c and Supplementary Fig. 4). This α-helical region forms relatively loose interactions with ECLs compared to the C-terminal tail of NPY that forms extensive interactions with the TM core. The cryo-EM map density was weaker toward the N-terminus of the helix, suggesting the flexibility of this region. Three independent 1-μs MD simulations using a model composed of full-length NPY, Y₁R (2–339), and G_{i1} protein showed that the angle of the helical axis of NPY to the membrane normal varied from 5° to 70° during the simulations (Supplementary Fig. 14). While the C-terminal tail of NPY maintained its binding pose during the simulations, the movement increased toward the N-terminus of the NPY helix, explaining the weak density of the N-terminal end of the NPY helix.

The density of side chains in the ECL2 region spanning residues 185–193 was unclear, but we identified that P183ᴱᶜᴸ²,

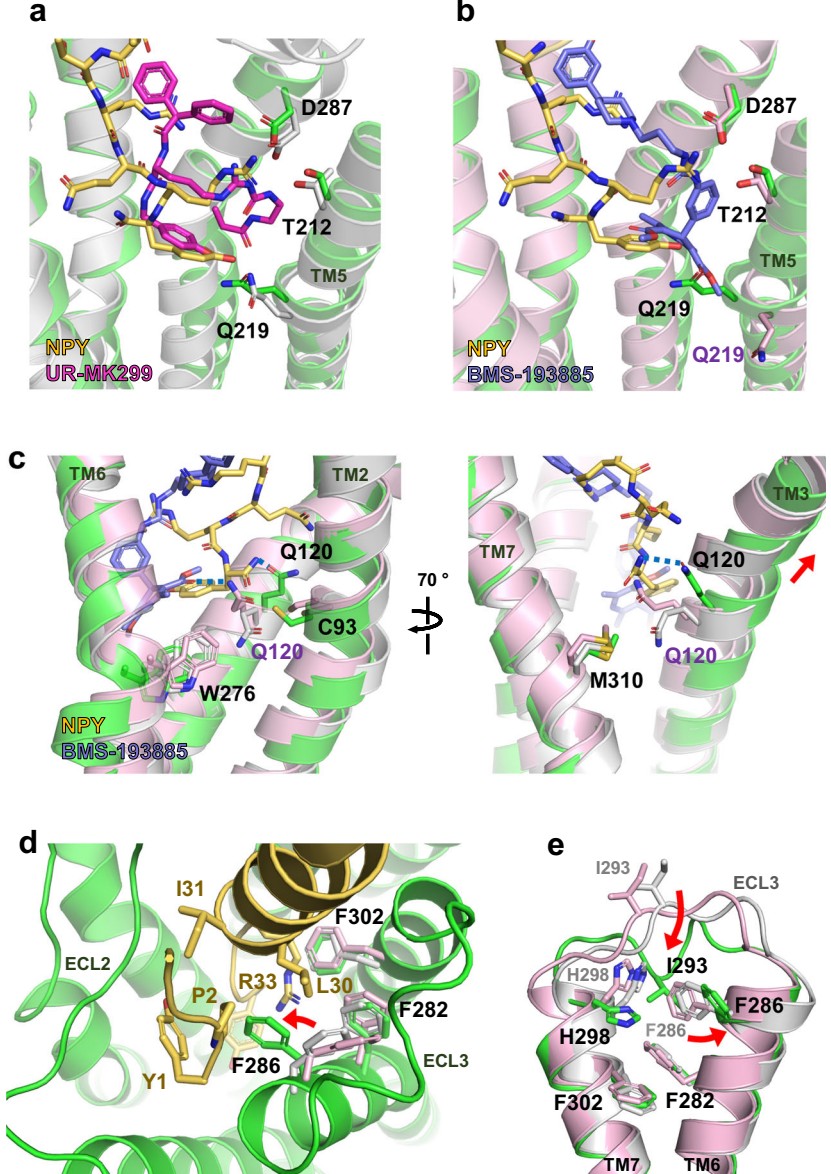

**Fig. 4 Comparison of binding mode between NPY and antagonists. a** Ligand-binding site occupied by UR-MK299 (PDB ID 5ZBQ, magenta) and **b** BMS-193885 (PDB ID 5ZBH, slate) is compared with the binding site for NPY (yellow). The side chains of Q219$^{5.46}$ are differently oriented in all three structures, whereas D287$^{6.59}$ and T212$^{5.39}$ maintain similar interactions. **c** Q120$^{3.32}$-mediated interactions in the antagonist-bound and NPY-bound structures are compared. In the NPY-bound active structure of Y$_1$R, TM3 including Q120$^{3.32}$ shifts upward and Q120$^{3.32}$ forms a polar interaction with the C-terminal amide of NPY through its upward-facing sidechain. **d** Upon NPY binding, F286$^{6.58}$ rotamer is changed to form the interactions with NPY R33, L30, and Y1. **e** Upon NPY binding, I293$^{ECL3}$ participates in a hydrophobic interaction network with F286$^{6.58}$, H298$^{7.31}$, and F302$^{7.35}$. UR-MK299-bound Y$_1$R is indicated in light gray, and BMS-193885-bound Y$_1$R is in pink. Red arrows represent positional changes of I293$^{ECL3}$ and F286$^{6.58}$ upon NPY binding.

F184$^{ECL2}$, and F199$^{ECL2}$ interacted with Y27, I28, and I31 of NPY, respectively (Fig. 5a). The F184$^{ECL2}$A and F199$^{ECL2}$A mutants reduce the potency of NPY by 38-fold and 2.3-fold, respectively (Supplementary Table 2 and Supplementary Figs. 15, 16, and 17). Although the F202$^{ECL2}$A mutation exhibited attenuated G protein signaling, F202$^{ECL2}$ did not interact with NPY. F202$^{ECL2}$ appears to be important for maintaining the structural integrity of the receptor by forming an aromatic network with F173$^{4.60}$, Y176$^{4.63}$, and Y211$^{5.38}$, as observed in both inactive and active structures (Supplementary Fig. 18).

Previously, it was proposed that the complete N-terminus of NPY is necessary for NPY signaling through Y$_1$R[10]. Consistently, our signaling assays demonstrated the decrease in the potency of NPY(3–36) and NPY(18–36) by 18-fold and 300-fold,

respectively, compared to full-length NPY (Fig. 5b), suggesting that the N-terminal residues of NPY are important for Y$_1$R binding. In our structure, NPY residues 1–5 were modeled by fitting these residues into a cryo-EM map (Fig. 1c and Supplementary Fig. 19). Among these five residues, the N-terminal residue Y1 was relatively well defined by the cryo-EM density. Y1 interacts with F199$^{ECL2}$, D200$^{ECL2}$, and R208$^{5.35}$ of Y$_1$R (Fig. 5c), all of which were demonstrated to be important for NPY signaling by previous or current mutational studies[27,28] (Supplementary Table 2 and Supplementary Figs. 15, 16, and 17).

The N-terminus of NPY does not enter the TM core like the C-terminus but is exposed to the solvent, suggesting its propensity to interact more dynamically with Y$_1$R. In agreement with our speculation, MD simulations show that the N-terminal

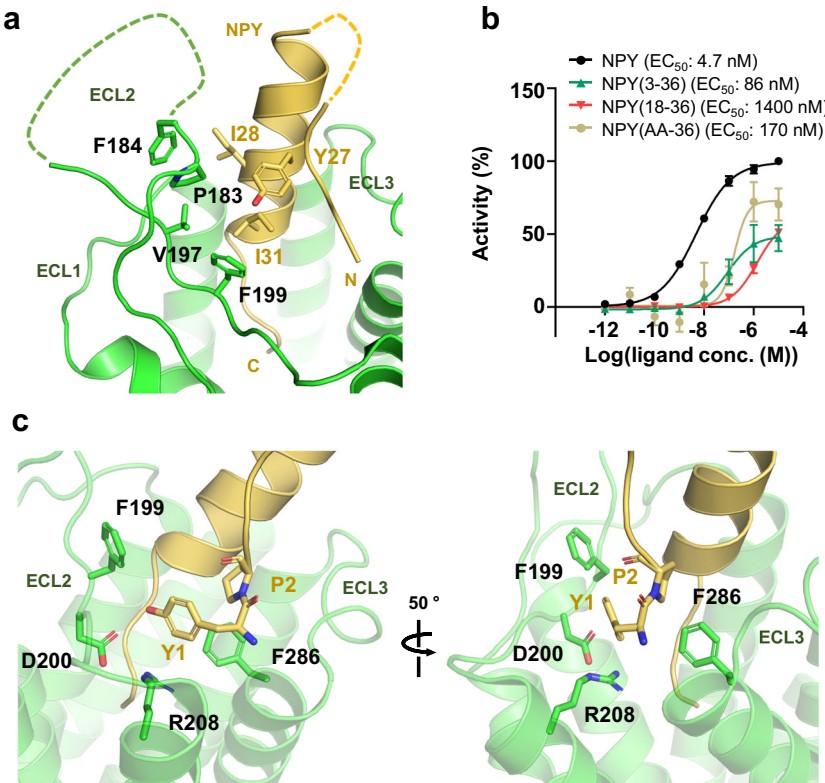

**Fig. 5 Binding of NPY N-terminus to Y₁R. a** The sticks show the residues participating in van der Waals interactions between NPY helix and ECL2 of Y₁R. Unresolved regions of NPY and ECL2 of Y₁R are shown with dashed lines. **b** The results of $Ca^{2+}$ assays performed with N-terminally truncated NPY peptides are shown. Removal of two N-terminal residues dramatically reduces NPY potency and efficacy. Symbol and error bar indicate the mean and S.E.M. (standard error of mean) of $n$ = three (NPY(3–36), NPY(18–36), and NPY(AA-36)) or $n$ = 17 (NPY) independent experiments, respectively. Calculated $EC_{50}$ values ($pEC_{50}$ ± SEM) are presented on each graph and are summarized in Supplementary Table 2. Source data are provided as a Source Data file. **c** Van der Waals interactions of Y1 and P2 of NPY with Y₁R are shown in two different orientations.

region of NPY exhibits relatively higher motions than the C-terminal tail (32–36) during 1-μs simulation. Moreover, the Y1-mediated interactions were broken and reformed in one of the replicates during the simulations (Supplementary Fig. 14). Collectively, our data show that the N-terminal region of NPY interacts dynamically with Y₁R. Our model structure represents a possible conformation of the NPY N-terminus, deduced from the cryo-EM map.

**Functional role of the N-terminal region of Y₁R.** Our cryo-EM map reveals a continuous density from the N-terminal end of TM1 toward the NPY ligand (Supplementary Fig. 6), suggesting an interaction between them. Previously, the N-terminal region of Y₁R, residues 21–32, was crosslinked to NPY in a photo-crosslinking experiment[10]. However, the low-resolution map at TM1 and the N-terminal region of Y₁R prevented us from unambiguously determining whether the N-terminal region of Y₁R directly interacted with NPY. Through signaling assays and structural analysis, it was previously proven that N-terminal residues were critical for peptide ligand binding and activation of other closely related peptide receptors, namely OX₂R and ET_BR[29–32].

To validate the importance of the N-terminal region of Y₁R for NPY signaling, we constructed two N-terminal deletion mutants of Y₁R, one in which the 25 N-terminal residues were deleted (Y₁RΔ25) and the other in which the 31 N-terminal residues were deleted (Y₁RΔ31), and performed BRET and $Ca^{2+}$ assays (Supplementary Table 2 and Supplementary Figs. 20, 21, and 22). Y₁RΔ25 behaves like wild-type Y₁R in recruiting G_{i1} by NPY

treatment, whereas Y₁RΔ31 showed attenuated response to NPY; this finding suggests that residues 26–31 are involved in NPY binding and thus in G_{i1} recruitment. In particular, the hydrophobic residues L26^N and F28^N in Y₁R were important for NPY signaling, as indicated by the 2.8-fold increase in $EC_{50}$ in the L28^N A/F28^N A mutant (Supplementary Table 2 and Supplementary Figs. 20, 21, and 22). MD simulations show that although the N-terminal region of Y₁R has high mobility, L26^N and F28^N are within the Cα distance of 9 and 13 Å, respectively, from Y21 in the helical region of NPY (Supplementary Fig. 23). Of note, Y21 of NPY is assumed to be a contact point based on the nearby extra cryo-EM density. In addition, the helical region of NPY was demonstrated to interact with ECL2 during the simulations, suggesting that ECL2 and the N-terminal region of Y₁R form dynamic interactions with NPY by reorienting themselves extensively to accommodate the NPY binding (Supplementary Fig. 23). Thus, we hypothesize that the helical region of NPY forms a tripartite interaction with ECL2 and the N-terminal region of Y₁R, both are shown to partially cover the ligand-binding pocket in the antagonist-bound Y₁R structures.

## Discussion

Several structures of the G protein-bound active state of class A GPCRs in complex with endogenous peptide agonists, such as NTS (8–13), OxB, and cholecystokinin-8 (CCK-8), have been reported[24,33,34]. Commonly, these peptide agonists have a C-terminal region that inserts into the receptor TM core and acts as a "message" domain (Supplementary Fig. 24)[35,36]. Similarly, in this study, we observed that the five NPY C-terminal residues in

the NPY-bound $Y_1R$ structure, forming an extended conformation, make extensive contact with residues in the TM core. In addition, our structure shows that the helical region and the N-terminal loop of NPY interact with $Y_1R$, although this interaction is much more dynamic and even transient, as indicated by the weaker density for this region in our cryo-EM map, as well as our MD simulations. This study presents a model candidate containing the five N-terminal residues of NPY, constructed based on our cryo-EM map. The cryo-EM density for Y1 of NPY is relatively well resolved, showing interaction with $F199^{ECL2}$, $D200^{ECL2}$, and $R208^{5.35}$ of $Y_1R$. Despite being a dynamic interaction, the NPY N-terminus is crucial for $G_i$ signaling, as demonstrated by reduced $G_i$ recruitment in the BRET assay and a 18-fold increase in $EC_{50}$ value in signaling assays after treatment with NPY(3–36). The NPY receptor has two other peptide ligands, PYY and PP. PYY is released in response to nutrient intake along the gut and is highly homologous to NPY with 67% sequence identity; its N-terminus starts with tyrosine, similar to NPY (Supplementary Fig. 25). However, the major circulating form of PYY is the cleaved form PYY(3–36), known to be selective for $Y_2R$[12,13,37]. Our signaling assays with NPY, PYY, NPY(3–36), and PYY(3–36) also show that $Y_1R$ has $EC_{50}$ values in the nanomolar range for NPY and PYY (4.7 and 6.1 nM, respectively) and 13–18-fold increased $EC_{50}$ values for NPY(3–36) and PYY(3–36) (86 nM and 77 nM, respectively) (Supplementary Fig. 26), suggesting that PYY would bind $Y_1R$ similarly to NPY if its N-terminus remains intact. On the contrary, PP is secreted in the pancreas and has 50% sequence identity with NPY (Supplementary Fig. 25). Reportedly, PP does not bind $Y_1R$ at all[38]. PP has A1 and P34 instead of Y1 and Q34, respectively; thus, the interactions of Y1 and Q34 as shown in our NPY-bound $Y_1R$ structure are important for receptor binding. Indeed, our signaling assay shows that $Y_1R$ has a 64-fold increased $EC_{50}$ value for PP compared to NPY (Supplementary Fig. 26). Among four subtypes of the NPY receptor, $Y_4R$ was activated in response to PP[38–40]. Interestingly, $Y_4R$ has Glu at the position of 6.58 ($E288^{6.58}$), instead of Phe as in Y1R (Supplementary Fig. 27), suggesting that $Y_4R$ would form a charged interaction network with nearby charged residues ($E203^{ECL2}$, $R211^{5.35}$, $T215^{5.39}$, $N285^{6.55}$, $E288^{6.58}$, $D289^{6.59}$) and basic residues of PP, R33 and R35 (Supplementary Fig. 28). We speculate that this extensive charged interaction network would provide sufficient interaction energy for $Y_4R$ to accommodate PP as well as NPY, which should be validated with experimental data.

Unlike the previously predicted NPY binding pose, our structure shows that the C-terminal amide of NPY points toward $Q120^{3.32}$ of $Y_1R$, and the Y36 side chain interacts with $Q219^{5.46}$. The importance of $Q120^{3.32}$ and $Q219^{5.46}$ for NPY binding and signaling is demonstrated by the approximately 2 and 4-fold increased $EC_{50}$ values measured using the mutants $Q120^{3.32}A$ and $Q219^{5.46}L$, respectively. Notably, the Y36 side chain occupies a position similar to the hydroxyphenyl ring of the antagonist UR-MK299[10]. Thus, in both active and inactive structures, $Q219^{5.46}$ forms a hydrogen bond with the hydroxyl group of the hydroxyphenyl ring in each ligand. In contrast, $Q120^{3.32}$ of $Y_1R$ forms a hydrogen bond with the C-terminal amide of NPY; however, it is not involved in antagonist binding. Notably, the conserved Gln residue at position 3.32 in $OX_2R$ ($Q134^{3.32}$) forms a hydrogen bond with the peptide agonist. Therefore, it was proposed to be a key residue in facilitating the transition to an active state of $OX_2R$ by a rotamer change to its upward-facing extended conformation[24]. This proposal appears to apply to the $Y_1R$ activation mechanism, as a similar rotamer change of $Q120^{3.32}$ is observed upon activation (Fig. 6).

Comparison of the inactive and active structures suggests the activation mechanism of $Y_1R$ upon NPY binding. In the

published antagonist-bound structures, the three phenylalanine residues $F282^{6.54}$, $F286^{6.58}$, and $F302^{7.35}$ located near the entrance to the ligand-binding pocket constitute a hydrophobic cluster with the antagonists UR-MK299 and BMS-193885. Upon NPY binding, R33 of NPY is inserted into this Phe network, causing a rotamer change in $F286^{6.58}$, disrupting the aromatic network. In association with the conformational change of ECL3, a new interaction network including Y1 and R33 of NPY and $F282^{6.54}$, $F286^{6.58}$, $F302^{7.35}$, $I293^{ECL3}$, and $H298^{7.31}$ of $Y_1R$ is formed, stabilizing the NPY-bound $Y_1R$ structure (Fig. 4e). At the bottom of the ligand-binding pocket, Y36 of NPY forms hydrophobic interaction with $I124^{3.36}$ through the phenyl group and polar interaction with $Q120^{3.32}$ through the amidated C-terminus, leading to a rotamer change of $Q120^{3.32}$. This is followed by a rotamer change of $I128^{3.40}$, which interacts with $I124^{3.36}$ and repacking of the side chains of $P223^{5.50}$, $F272^{6.44}$, and $W276^{6.48}$. A series of these changes upon NPY binding pulls TM3 upward and causes outward movement of the cytoplasmic region of TM6 (Fig. 6).

One of the key NPY-interacting residues, $F286^{6.58}$, is not conserved and replaced by valine in $Y_2R$ and glutamate in $Y_4R$, both of which cannot form π–cation interaction with R33 as $F286^{6.58}$ does. The difference in the Phe network may explain the $Y_1R$ selectivity of the two antagonists used to determine the inactive $Y_1R$ structures; additionally, it suggests that $Y_1R$, $Y_2R$, and $Y_4R$ may have different interaction networks with NPY. In addition to $F286^{6.58}$, $Q219^{5.46}$, which forms a hydrogen bond with Y36, is replaced with $L227^{5.46}$ in $Y_2R$, and $H298^{7.31}$, which forms an interaction network with $F282^{6.54}$, $F286^{6.58}$, and $F302^{7.35}$ is replaced with $G300^{7.31}$ in $Y_4R$. It should also be noted that $F286^{6.58}$ is located within a distance of 4–5 Å to Y1 and P2 of NPY. Further details would only be explained by investigating the NPY-bound $Y_2R$ and $Y_4R$ structures in the future. However, the vicinity of the residue and the importance of the Phe network suggest that $F286^{6.58}$ may contribute to the difference between $Y_1R$ and $Y_2R$ in the need for the complete N-terminus to elicit a full response.

## Methods

**Expression and purification of $Y_1R$.** Wild-type human $Y_1R$ (2–384) with a FLAG tag at its N-terminus as well as eGFP and a $His_8$ tag at its C-terminus, cleavable by HRV 3C protease, was expressed in *Spodoptera frugiperda* (Sf9) insect cells using the Bac-to-Bac system (Invitrogen). Cells were harvested 48 h after infection and lysed by repeated dounce homogenization with lysis buffer (20 mM HEPES pH 8.0, 150 mM NaCl, and 1 mM EDTA) supplemented with phenylmethylsulfonyl fluoride (PMSF), benzamidine, and leupeptin. $Y_1R$ was extracted from the cell membrane using a solubilization buffer consisting of 20 mM HEPES pH 8.0, 150 mM NaCl, 1% (w/v) n-dodecyl-β-D-maltoside (DDM), 0.1% (w/v) cholesterol hemisuccinate (CHS), PMSF, benzamidine, and leupeptin. After centrifugation, the supernatant was incubated with Ni-NTA resin for 1 h at 4 °C. After column washing with wash buffer (20 mM HEPES pH 8.0, 150 mM NaCl, 20 mM imidazole, 0.05% DDM, and 0.005% CHS), the bound protein was eluted with 300 mM imidazole and subsequently loaded onto anti-FLAG M1 agarose resin (Sigma Aldrich) in the presence of 2 mM $CaCl_2$. After column washing and on-column exchange of detergent from DDM to glyco-diosgenin (GDN), the bound protein was eluted with M1 elution buffer (20 mM HEPES pH 8.0, 150 mM NaCl, 0.01% GDN, 0.1 mg/ml FLAG peptide, and 4 mM EDTA). Heterogeneous glycosylation was removed by PNGase F (NEB), and the eGFP was cleaved by the HRV 3C protease (homemade). $Y_1R$ was further purified using a Superdex 200 10/300 gel filtration column (Cytiva) pre-equilibrated with a buffer containing 20 mM HEPES pH 8.0, 150 mM NaCl, and 0.01% GDN. Freshly purified $Y_1R$ was used to form a complex with $G_{i1}$ heterotrimer.

**Purification of the G protein.** Human $Gα_{i1}$ with a $His_6$ tag at its N-terminus was expressed in *Escherichia coli* (*E. coli*) Rosetta (DE3) cells. Protein expression was induced with 0.5 mM isopropyl β-D-1-thiogalactopyranoside (IPTG), and the cells were harvested after incubation at 25 °C overnight. Cells were lysed with Emulsiflex C3 (Avestin), and the cleared lysate after centrifugation was loaded onto the Ni-NTA column. The column was washed with wash buffer (20 mM Tris-HCl pH 8.5, 150 mM NaCl, 30 mM imidazole), and the bound protein was eluted with elution buffer (20 mM Tris-HCl pH 8.5, 150 mM NaCl, and 300 mM imidazole). A $His_6$

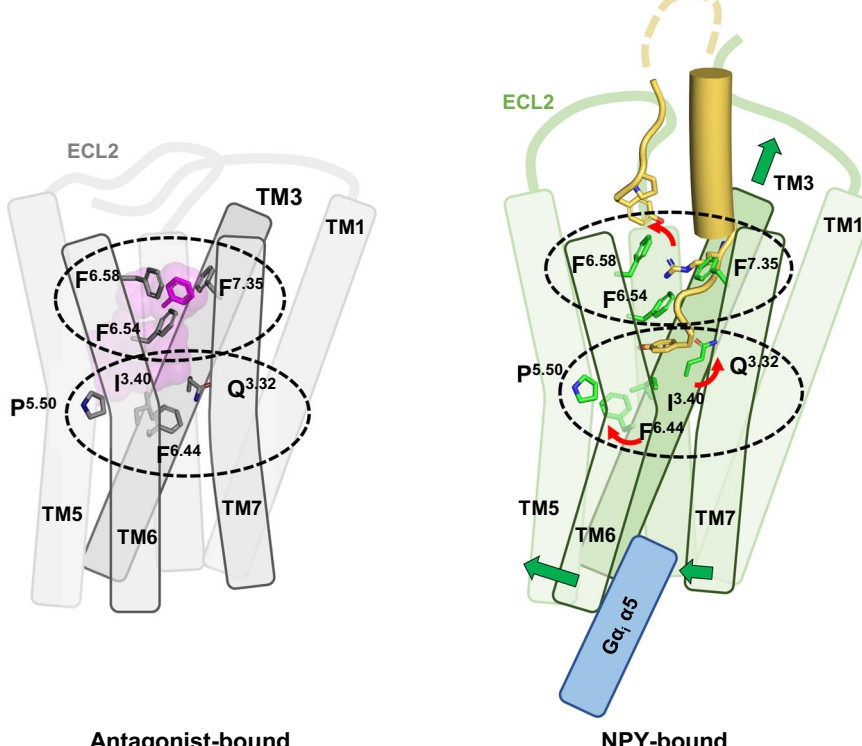

**Fig. 6 Mechanism of Y$_1$R activation by NPY binding.** Schematic figures representing antagonist-bound (left) and NPY-bound (right) Y$_1$R are shown. Two key regions for Y$_1$R activation are indicated by dashed ellipses. The three phenylalanine residues at TM6 and TM7, and Q120$^{3.32}$, I128$^{3.40}$, P223$^{5.50}$, and F272$^{6.44}$ in the connector region are shown in stick representation. The three phenylalanine residues participate in antagonist binding through π–π interactions, but these interactions are rearranged upon NPY binding. F286$^{6.58}$ flips and forms a new interaction network with Y1 and R33 of NPY. Since F286$^{6.58}$ is not conserved in Y$_2$R, F286$^{6.58}$-mediated interactions are Y$_1$R-selective. Q120$^{3.32}$, I128$^{3.40}$ and F272$^{6.44}$ show rotamer changes upon NPY binding. A series of these changes upon NPY binding pulls TM3 upward and causes outward movement of the cytoplasmic region of TM6.

tag was cleaved by TEV protease (homemade) treatment at 4 °C overnight. The Hitrap Q column (Cytiva) and a gel filtration column were used for further purification. Purified Gα$_{i1}$ dissolved in a solution composed of 20 mM Tris-HCl pH 8.5, 50 mM NaCl, 1 mM MgCl$_2$, and 10 μM guanosine diphosphate (GDP) was concentrated, snap frozen in liquid nitrogen, and stored at −80 °C until use.

Human Gβ$_1$ with a His$_6$ tag at its N-terminus and Gγ$_2$ with a C68S mutation were co-expressed in Sf9 insect cells. Cells were harvested 72 h after incubation and lysed with lysis buffer (20 mM Tris-HCl pH 8.5, 0.1 mM Tris (2-carboxyethyl) phosphine hydrochloride (TCEP), and protease inhibitors). After centrifugation, the cleared supernatant was loaded onto an Ni-NTA column. The bound protein was eluted with lysis buffer supplemented with 300 mM imidazole after washing with lysis buffer supplemented with 20 mM imidazole. The His$_6$ tag was cleaved with HRV 3C protease, and the Gβγ complex was further purified using a Hitrap Q column.

For G protein heterotrimer formation, purified Gα$_{i1}$ and Gβγ were mixed in a 1.1:1 molar ratio with excess MgCl$_2$ and GDP. After an hour of incubation on ice, the Gα$_i$βγ complex was purified using a gel filtration column equilibrated with a solution composed of 20 mM HEPES pH 7.5, 150 mM NaCl, 1 mM MgCl$_2$, and 10 μM GDP. The purified Gα$_i$βγ complex was concentrated, snap frozen in liquid nitrogen, and stored at −80 °C until use.

We confirmed that Gα$_{i1}$ produced from *E. coli* is still functional using GTP turnover assay (Supplementary Fig. 29). For comparison, we purified G$_{i1}$ heterotrimer produced from Sf9 insect cells, as previously described[22].

**Purification of the scFv16.** The scFv16 construct was kindly provided by Dr. Kobilka (Stanford University). Purification of scFv16 was performed by the previously described method with slight modifications[19]. First, scFv16 with a His$_6$ tag at its C-terminus was expressed in *Trichoplusia ni* (Hi5) cells. After cell harvesting, the supernatant containing secreted scFv16 was incubated with Ni-NTA resin at 4 °C for 2 h. The resin was washed with buffer (20 mM HEPES pH 7.0, and 150 mM NaCl, 30 mM imidazole), and the bound protein was eluted with the buffer supplemented with an additional 300 mM imidazole. Further purification was performed using a gel filtration column pre-equilibrated with a solution composed of 20 mM HEPES pH 7.0, and 150 mM NaCl. Purified scFv16 was concentrated, snap frozen in liquid nitrogen, and stored at −80 °C until use.

**Purification of the NPY–Y$_1$R–G$_{i1}$-scFv16 complex.** Purified Y$_1$R, G$_{i1}$ heterotrimer, and scFv16 were mixed at a molar ratio of 1:1.1:1.2 in the presence of excess NPY (GL Biochem, Shanghai, China) and incubated overnight at 4 °C with apyrase (NEB, MA, USA). The sample was then loaded onto a gel filtration column pre-equilibrated with a solution composed of 20 mM HEPES pH 8.0, 150 mM NaCl, 0.01% GDN, and 2 μM NPY. The purified NPY–Y$_1$R–G$_{i1}$–scFv16 complex was concentrated to 10 mg/ml and used for cryo-EM grid preparation.

**Cryo-EM grid preparation and data collection.** An aliquot (3.5 μl) of purified NPY–Y$_1$R–G$_{i1}$–scFv16 complex was applied onto a glow-discharged holey carbon grid (Quantifoil R1.2/1.3, 300 mesh). The grids were blotted for 5 s at 12 °C and 100% humidity and plunge-frozen in liquid ethane using a Vitrobot Mark IV (Thermo Fisher Scientific, USA) at Center for Macromolecular and Cell imaging of Seoul National University (SNU CMCI). Grids were initially screened with the FEI Glacios (Thermo Fisher Scientific, USA) at SNU CMCI, equipped with a Falcon 4 detector. Images were acquired on a 300-kV Titan Krios (Thermo Fisher Scientific, USA) at Korea Basic Science Institute, equipped with a Falcon 3EC direct electron detector. Movies were recorded in counting mode at a magnification of ×161,850 (corresponding to a calibrated pixel size of 0.865 Å) and a defocus range of −1.25 to −2.75 μm. A total of 4965 movies were collected, each comprising 40 frames, with a total dose of 40 electrons per Å$^2$. A detailed description of the cryo-EM data collection parameters is provided in Supplementary Table 1.

**Three-dimensional (3D) reconstruction of NPY–Y$_1$R-G$_{i1}$–scFv16 complex.** Image stack preprocessing was performed using CryoSPARC v. 3.1 (Structura Biotechnology)[41]. All movies subjected to beam-induced motion correction using patched-motion correction and contrast transfer function (CTF) parameters for each non-dose-weighted micrograph were determined by patch CTF estimation. After initial particle picking and two-dimensional classification, selected good particles were used for Topaz training[42]. Topaz picking particles (1,300,000 particles) were extracted and subjected to three rounds of heterogeneous refinement. Further heterogeneity classifications were performed by 3D-variability analysis (3DVA)[43], focusing on the NPY–Y$_1$R–G$_{i1}$ complex without micelles, Gα$_i$ AHD, and scFv16 by masking. Clusters with well-resolved density were obtained and used for final map reconstruction by non-uniform refinement[44]. Maps for this

processing have a global nominal resolution of 3.2 Å, based on gold-standard Fourier shell correlation using the 0.143 criteria. To improve the map quality of NPY–Y$_1$R–G$_{i1}$ complex, we performed the local refinement focusing on the NPY–Y$_1$R and G$_{i1}$–scFv16 in cryoSPARC v3.2. These local refinements generated the maps at a global nominal resolution of 3.6 Å and 3.1 Å, respectively. Cryo-EM density for ECL2 (176–183, 194–203) and NPY (1–5, 20–36) was clearly observed in the refined map with a mask on NPY–Y$_1$R. These two local refined maps were combined using "vop maximum" command in UCSF chimera to represent and analyze the NPY–Y$_1$R–G$_{i1}$ complex[45]. The combined map is represented in Fig. 1a. The local resolution was determined using the cryoSPARC local resolution estimation. Local sharpening was performed by LocSpiral to trace the Y$_1$R N-terminus (beyond L35$^N$) and ECL2 region (Supplementary Fig. 6)[46]. No artifacts were observed when compared to the global sharpening map in the other regions.

**Model building and refinement.** The initial model was obtained by rigid-body-fitting of the structure of inactive Y$_1$R (PDB ID 5ZBH)[10] and Gα$_i$βγ and scFv16 from the NTSR1 complex structure (PDB ID 6OS9)[22]. This initial model was then subjected to iterative rounds of manual rebuilding with COOT and refinement with PHENIX[47,48]. The geometry of the refined structure was evaluated using the MolProbity[49]. The final model consisting of NPY, Y$_1$R, Gα$_i$, Gβ, Gγ, and scFv16 was deposited in the PDB with PDB code 7VGX, and the electron density map was deposited in the EMDB with ID EMD-31979. The refinement statistics are presented in Supplementary Table 1. All molecular graphic figures were prepared using the UCSF Chimera, UCSF ChimeraX, and PyMol v2.4.0[45,50,51].

**GTP turnover assay.** GTP turnover assay was performed using GTPase-Glo assay kit (Promega)[22]. Purified Y$_1$R was incubated with NPY for 40 min at room temperature. NPY-bound Y$_1$R (4 μM) was mixed with 1 μM G$_{i1}$ heterotrimer (containing Gα$_i$ produced from *E. coli* or Sf9 cells) in an assay buffer consisting of 20 mM HEPES pH 8.0, 150 mM NaCl, 0.03% DDM, 0.003% CHS, 100 μM TCEP, 5 μM GDP, and 2.5 μM GTP. After incubation for 3 h, reconstituted GTPase-Glo reagent was added to the sample and incubated for 30 min at room temperature. Then, detection reagent was mixed and incubated for 10 min at room temperature. Luminescence was measured using a FlexStation 3 multi-mode microplate reader (Molecular Devices) and data were analyzed by GraphPad Prism 9.2.0.

**ELISA-based surface expression assay.** HEK293T cells were transfected with each expression plasmid of Y$_1$R mutants and wild-type. After 48 h of incubation, 4% paraformaldehyde (PFA, T&I) was treated for fixation and washed with 1× PBS. After 30 min of incubation with blocking solution (2.5% bovine serum albumin, Bovogen), rabbit anti-FLAG antibody (Cell signaling Technology, 1:1000 dilution) was treated for staining and anti-rabbit HRP antibody (Enzo Life Sciences, 1:1000 dilution) was used for detection. After incubation with HRP antibody, TMB solution (Thermofisher scientific) was added to each well, incubated until blue color was observed. Further reaction was blocked by adding 2 M HCl. The absorbance was detected at 450 nm with FlexStation 3 multi-mode microplate reader. Normalization was carried out by removing TMB substrate solution from the wells and adding Janus Green solution (0.2% w/v, TCL). Further elimination of excess stain was done by washing with milli Q water and adding 0.5 M HCl. Absorbance was read at 595 nm. Normalized expression level of the receptor at the cell surface was calculated by the ratio of the absorbance at 450 and 595 nm ($A_{450}/A_{595}$). The graphs were plotted using GraphPad Prism 9.2.0.

**BRET assay.** HEK293T cells were co-transfected with Gα$_i$-Rluc, Gβ, Gγ, and Y$_1$R-eYFP constructs at a 1:1:1:5 ratio. Forty-eight hours of post-transfection, cells were detached with PBS supplemented with 20 mM EDTA and evenly spread into white 96-well microplates (SPL). For ligand-induced conditions, various concentrations of NPY were incubated with each Y$_1$R mutant-transfected group, and Coelenterazine h was added to each well to a final concentration of 5 μM. All BRET data were collected using a Mithras LB940 instrument (Berthold), and graphs were plotted using GraphPad Prism 9.2.0.

**Ca$^{2+}$ signaling assay.** For the ligand-induced Ca$^{2+}$ assay, HEK293T cells were seeded on 96-well black wall/clear bottom microplate (SPL) in Dulbecco's modified Eagle's medium (Biowest) containing 10% fetal bovine serum (Biowest) and antibiotic-antimycotic (Gibco) 24 h before plasmid transfection. After transfection with the plasmid containing Y$_1$R constructs, Gα$_{Δ6qi4myr}$[52] Gβ, and Gγ in a 3:1:1:1 ratio, followed by 48 h incubation, cells were stained with Cal-520 (AAT Bioquest, Inc.) in assay buffer (HBSS, 0.1% BSA, 20 mM HEPES pH 7.4). After 2 h, Cal-520-stained cells were washed three times with assay buffer. Intracellular Ca$^{2+}$ influx was measured at Ex/Em = 490/525 nm using the FlexStation 3 multi-mode microplate reader (Molecular Devices). After 30 s of baseline, the ligand was injected to achieve the final concentration. Log (concentration)-response curves, used to estimate EC$_{50}$, were calculated using GraphPad Prism 9.2.0, by fitting an agonist response curve with a variant slope to the normalized response data.

**Calculation of the ligand-binding pocket volume.** First, to define the interior of the TM bundle, the receptor structure was aligned along the *z*-axis by superimposing a pre-aligned GPCR structure from the OPM database[53]. Thereafter, a 3D grid was constructed with equispaced points covering the receptor structure. The centers of the TM helices for each discrete *z*-axis value and the lines connecting the neighboring centers were defined as the lateral boundaries. For helices without a defined center point for a given *z*-axis value, the (*x*,*y*) coordinate of the nearest point was used instead. Next, the upper and lower boundaries of the ligand-binding pocket were defined. The *z*-axis coordinate of the Cα atom of W$^{6.48}$, a toggle switch residue, was defined as the lower boundary. TM residues closest to the extracellular region were used to define the upper boundary of the pocket volume. Finally, after removing the grid points causing clashes with protein atoms, the cavity volume was calculated from the number of grid points inside the defined boundaries. The python code for calculating the solvent accessible volume of the ligand-binding pocket is available at https://github.com/seoklab/GPCR_binding_cavity_volume_calculation.

**Y$_4$R-PP homology modeling.** The homology model of PP-bound Y$_4$R was prepared by template-based modeling protocol, GalaxyTBM, using the current NPY-bound Y$_1$R structure as a template[54]. The initial model was constructed by threading the target sequence on the template structure, followed by energy optimization and additional structure sampling for unreliable local regions. Physics-based optimization method GalaxyRefineComplex refined template-based models[55]. A scoring function optimized for GPCR structure prediction in Galaxy7TM was used[56].

**MD simulation.** Missing residues were added to the structure model of the NPY–Y$_1$R–G$_{i1}$ complex to build an initial model for MD simulation. The AHD domain of G$_{i1}$ was added by aligning the previously reported structure[57], and the missing residues of Y$_1$R in the N-terminus (25–34), ECL2 (184–193), ICL2 (247–252), and C-terminus (330–339) were built based on the map with a mask on NPY–Y$_1$R at a lower threshold. The rest of the N-terminus (2–24) was extended randomly in a position that did not collide with the existing structure.

This study used the CHARMM36(m) force field for proteins and lipids[58–60]. The TIP3P water model was utilized along with 0.15 M NaCl solution[61]. Three independent MD simulations were performed for each system to obtain better sampling and check the convergence. Periodic boundary conditions (PBCs) were employed in all simulations. The van der Waals interactions were smoothly switched off over 10–12 Å by a force-based switching function and the long-range electrostatic interactions were calculated using the particle-mesh Ewald method with a mesh size of ~1 Å[62]. All simulations were performed using the inputs generated by CHARMM-GUI and GROMACS 2018.6 for both equilibration and production with the LINCS algorithm[63–67]. The temperature was maintained using a Nosé-Hoover temperature coupling method with a τ$_t$ of 1 ps[68]. For pressure coupling (1 bar), the semi-isotropic Parrinello–Rahman method with a τ$_p$ of 5 ps and compressibility of $4.5 \times 10^{-5}$ bar$^{-1}$ was used[69]. The constant particle number, volume, and temperature (NVT) dynamics were first applied with a 1-fs time step for 250 ps during the equilibration run. Subsequently, the constant particle number, pressure, and temperature (NPT) ensemble was applied with a 1 fs time step (for 2 ns) and with a 2 fs time step (for 18 ns). During the equilibration, positional and dihedral restraint potentials were applied, and their force constants were gradually reduced. The production run was performed with a 4 fs time step using the hydrogen mass repartitioning technique without any restraint potential[70]. Each system ran about 25 ns/day with 512 CPU cores on NURION in the Korea Institute of Science and Technology Information.

**Reporting summary.** Further information on research design is available in the Nature Research Reporting Summary linked to this article.

## Data availability

Additional data supporting the findings of this work are available as the Supplementary Information, Supplementary Data and Source Data files. The atomic model has been deposited in the Protein Data Bank under accession code 7VGX and the cryo-EM density maps have been deposited in the Electron Microscopy Data Bank under accession code EMD-31979. The inactive Y$_1$R (PDB ID 5ZBH), Gα$_{i1}$βγ and scFv16 from the NTSR1 complex structure (PDB ID 6OS9) were used as an initial template to build the NPY–Y$_1$R–G$_{i1}$ protein model. Structural models used in data analysis were accessed from the Protein Data Bank under the accession codes 5ZBQ (UR-MK299 and Y$_1$R), 6DDE (μOR, G$_i$), 7L1U (OxB and OX$_2$R), 5GLH (ET1 and ET$_B$R), 7L0Q (NTS and NTSR1), and 7MBX (CCK-8 and CCK1R). Source data are provided with this paper.

## Code availability

The python code for calculating the solvent accessible volume of the ligand-binding pocket is available at https://github.com/seoklab/GPCR_binding_cavity_volume_calculation and as a Source Data file.

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

## Acknowledgements

This work was supported by the Creative-Pioneering Researchers Program through Seoul National University (H.-J.C.), the National Research Foundation of Korea funded by the Korean government (NRF-2020R1A2C2003783 and NRF-2019M3E5D6063903 to H.-J.C.), and National Science Foundation, USA (MCB-2111728 to W.I.). This work was also supported by Korea Basic Science Institute grant (2020R1A6C101A183 to H.-J.C.) and the Korea Basic Science Institute under the R&D program (Project No. C140440 to H.-J.C. and H.J.) supervised by the Ministry of Science and ICT. We thank Dr. Brian Kobilka (Stanford University) for providing the scFv16 construct, Chul Won Choi and Jungnam Bae (Seoul National University) for supporting grid preparation, Junsun Park and Dr. Soung-Hun Roh (Seoul National University) for providing materials and methods for grid screening. We also thank Dr. Bum Han Ryu (Institute for Basic Science (IBS), Korea) and Dr. Jin Seok Choi (KAIST Analysis Center for Research Advancement (KARA), Korea) for supporting grid screening at IBS and KARA cryo-EM facilities, respectively, and Global Science experimental Data hub Center (GSDC) at Korea Institute of Science and Technology Information (KISTI) and the data analysis hub, Olaf at the IBS Research Solution Center for computing resources.

## Author contributions

C.P. and J.K. contributed equally to this work. C.P. purified the NPY–$Y_1$R–$G_{i1}$–scFv16 complex. J.K. collected and processed cryo-EM data with help from H.J. and I.B. C.P. and J.K. determined and refined the complex structure. S.B.K. performed cell-based assays including surface ELISA, BRET assays, and $Ca^{2+}$ assays with help from J.K., S.A.K., and T.-Y.Y. Y.K.C. and W.I. performed MD simulations and H.W. and C.S. calculated the volume of ligand-binding pockets and generated a homology model of PP-bound $Y_4$R. H.K. purified $G_{i1}$ heterotrimer and scFv16. H.-J.C. conceived and directed the study. C.P., J.K., I.B., and H.-J.C. wrote the manuscript with contributions from all authors.

## Competing interests

The authors declare no competing interests.
