## [Peer Review File · Nature Communications]

Structural basis of neuropeptide Y signaling through Y1 receptorREVIEWER COMMENTS

Reviewer #1 (Remarks to the Author):

This manuscript by Park et al reports a cryo-EM structure of the Neuropeptide Y (NPY) receptor 1 in complex with NPY and Gi. NPY receptors are interesting neuropeptide receptors in drug development due to their roles in the stimulation of food intake and the modulation of multiple functional aspects of the CNS. The structure reported in this paper represents the first structure of an NPY receptor signaling complex. The results revealed an interesting binding mode of NPY, in which both N- and C-terminal segments are involved in the receptor binding. Comparison to the antagonist-bound inactive structures of Y1R allowed the authors to define critical molecular features in the activation of Y1R. The molecular mechanism underlying the ligand selectivity of NPY receptors is also discussed. In particular, the additional binding of the N-terminal region of NPY to Y1R is very interesting. To the best of my knowledge, similar binding modes have not been reported for other neuropeptides. There are also rich mutagenesis data and MD simulation data to support the structural findings. Overall, the data quality is high. The quality of cryo-EM maps is sufficient. The data interpretation is reasonable. The figures are all nicely prepared and clear.

Minor comments:

1. The authors produced their Gi alpha subunit from E.coli, which is unusual for the structural characterization of GPCR signaling complexes. I would assume that the purified Gialpha is not lipidated. It will be helpful to prove that the Gi heterotrimer with an unmodified Gialpha is still functional using GTPγS binding or GTP turnover assays, or by biochemical data showing that the binding of purified Gi to Y1R is agonist dependent.
2. The binding of alpha5 of Gialpha to Y1R seems a bit different from that in other Gi-coupled GPCR structures. It may be helpful to compare the binding modes of Gi in the structures with Y1R and other neuropeptide GPCRs including NTSR1 and mu-opioid receptor.
3. The figure numbering needs to be revised. There is no Figure 1d.
4. Based on my understanding, all the MD simulation data shown in Supplementary Figures 12, 18, and 23 were from the same three repeated runs. It may be less confusing (at least to me) to put them in one figure as different panels.

5. The section "Difference in binding mode between antagonists and NPY C-terminal segment" may need more discussion of receptor activation mechanism, e.g. how the differences in the binding of antagonists and NPY lead to receptor activation. I would also suggest the authors revise the title of this section to indicate that there is a discussion of receptor activation mechanism, not just a description of differences in the binding of several different ligands.

6. The authors need to have a clearer definition of the N-terminal loop region, the helical region, and the C-terminal tail of NPY. It is quite confusing that the authors used these terms loosely in the discussion. For example, the authors stated that "the N-terminal region of Y1R, residues 21–32, was crosslinked to NPY in a photo-crosslinking experiment", while 21-32 is the helical region. When the authors talked about the N-terminal region, I am not sure if they referred to the N-terminal loop region (Y1-P5), or the loop region with the un-modeled region, or the entire region Y1-I31 including the helical region. I also recommend including clear terms for the three different regions of NPY in Figure 1C.

7. Is there any data showing that the extreme N-terminal region of NPY (Y1-P5) is important for the action of NPY? Why does it fold back to extend towards the receptor instead of sticking away from the receptor like other neuropeptides?

Reviewer #2 (Remarks to the Author):

*** Summary of the research and overall impression ***

Park and co-workers report the first experimental structure of the Y1 receptor (Y1R) in an active conformation bound to the natural agonist peptide neuropeptide Y (NPY) and heterotrimeric Gi determined by cryo-EM. The structure shows that the C-terminal portion of NPY adopts an extended conformation and binds deep within the core of Y1R. The N-terminal regions of the peptide are less well resolved, but cryo-EM combined with mutagenesis and MD simulations suggests dynamic interactions of this region with the extracellular areas of the receptor. The structure rationalizes mutagenesis results presented in this manuscript as well as previously published mutagenesis and NMR studies and reveals the peptide-receptor interactions important for receptor activation.

The manuscript is very interesting and adds important insight into how natural peptide agonists activate their cognate GPCRs. The research is of very high quality and the manuscript is logically structured. However, the clarity of the manuscript should be enhanced by editing for language and writing quality

to improve flow and readability. In addition, I suggest the authors address the following issues before publication.

Specific areas for improvement

*** Major issues ***

1. On p. 6, lines 1-5, the authors compare the solvent-accessible volumes of inactive-state and active-state Y1R. The relatively modest conformational changes in the extracellular ends of a subset of the TM helices described in the text and shown in Fig. 2a do not appear to explain the large increase in volume stated in the text (414 \AA^3 vs 859 \AA^3). The authors should verify and show more clearly where this 2-fold increase in volume originates from. Have unmodeled residues/sidechains been taken into account? Also, the volumes given in the main text do not match those given in Supplementary Fig. 5. Finally, the methods section on the calculation of the binding site volumes (p. 21) lacks information on what software was used.

2. The authors should expand their discussion of selectivity. The molecular basis of selectivity of the three related peptide agonists NPY, PYY, and PP is briefly discussed on p. 13. According to the work by Cabrele et al. cited in the manuscript (ref. 37), PP does not activate Y1R at all or only with very low potency, but instead is selective for Y4R and to a lesser degree Y5R. Conversely, NPY is a potent agonist of all four neuropeptide Y receptors. Based on the structure of NPY-bound Y1R presented by the authors and amino-acid differences in the binding sites of Y1R and Y4R, is it possible to explain the observed selectivity and in particular the high potency of PP on Y4R, perhaps with the help of homology modeling?

3. The proposed activation mechanism is an important finding of this work, but is currently only described in the legend of Fig. 6. The mechanism should be summarized in the discussion section of the main text.

*** Minor issues ***

1. The information presented in Supplementary Fig. 26 could be enhanced by coloring the peptide sequences by conservation.

2. p. 6, line 7-10: "...extracellular loops ... and the N-terminal region ... reorient themselves extensively...". This statement should be supported with a figure.

3. p. 7, last line: "A similar acidic ligand-binding pocket is observed in OX2R,...". This is difficult to discern from Supplementary Fig. 11b. Given that the C-terminal segment of OxB is mostly composed of hydrophobic residues rather than basic sidechains as in NPY, this may also not be relevant. Please comment.

4. p. 8, line 9: "...the phenyl ring of Y36 forms an intramolecular interaction with R35 of NPY,...". This is not obvious in Fig. 3a the authors refer to. The authors may want to consider adding another figure.

5. p. 13, lines 9 and 10: The sentence "This study presents...based on our cryo-EM map" is unclear and needs rephrasing.

*** Other points ***

1. The adjectives C-terminal and N-terminal are used instead of the nouns C-terminus and N-terminus throughout the manuscript. Also, the authors should settle on the use of carboxyl terminus or C-terminus (amino-terminus or N-terminus).

2. p. 2, line 6: The sentence "The extended conformation..." does not make sense as it stands. It is not the conformation that binds in the receptor core, but the peptide.

3. On several occasions, the authors refer to "Fig. 1d", e.g. on p. 7, line 6. No panel d is included in Fig. 1. Should this read "Fig. 1c"?

4. p.5, line 13: "scFv16" should be introduced properly and a reference included.

5. p. 5, line 15: "GDN" should be defined.

Reviewer #3 (Remarks to the Author):

The paper by Choi and coworkers describes their cryo-electron microscopy study of the complex between the peptide NPY, the Y1 receptor, and Gi protein, together with ScFv16. The study is conducted at a high scientific level.

The authors point to the very significant potential of NPY, PYY3-36, and other peptides in this family, as potential therapeutics for the treatment of obesity, cancer and other diseases.

I am not a structural biologist but assessed the paper from a medicinal chemistry point of view. While some of their findings are in agreement with past reports on NPY-Y1R interactions, their study also provides new and detailed insights on how the peptide NPY binds the Y1 receptor. This will be very

interesting for peptide medicinal chemists, both when working with NPY and when working with peptides from other, related families.

There are small mistakes in references 30 and 53.

We thank the reviewers for their suggestions and comments.

We greatly appreciate their time and efforts in reviewing our manuscript. We believe that our revised manuscript has benefited from their insightful suggestions.

We have addressed every point raised by the reviewers in detailed point-by-point responses (written in blue),

REVIEWER COMMENTS

Reviewer #1 (Remarks to the Author):

This manuscript by Park et al reports a cryo-EM structure of the Neuropeptide Y (NPY) receptor 1 in complex with NPY and Gi. NPY receptors are interesting neuropeptide receptors in drug development due to their roles in the stimulation of food intake and the modulation of multiple functional aspects of the CNS. The structure reported in this paper represents the first structure of an NPY receptor signaling complex. The results revealed an interesting binding mode of NPY, in which both N- and C-terminal segments are involved in the receptor binding. Comparison to the antagonist-bound inactive structures of Y1R allowed the authors to define critical molecular features in the activation of Y1R. The molecular mechanism underlying the ligand selectivity of NPY receptors is also discussed. In particular, the additional binding of the N-terminal region of NPY to Y1R is very interesting. To the best of my knowledge, similar binding modes have not been reported for other neuropeptides. There are also rich mutagenesis data and MD simulation data to support the structural findings. Overall, the data quality is high. The quality of cryo-EM maps is sufficient. The data interpretation is reasonable. The figures are all nicely prepared and clear.

We appreciate the reviewer's comments on the originality of our research findings.

Minor comments:

1. The authors produced their Gi alpha subunit from E.coli, which is unusual for the structural characterization of GPCR signaling complexes. I would assume that the purified Gialpha is not lipidated. It will be helpful to prove that the Gi heterotrimer with an unmodified Gialpha is still functional using GTPγS binding or GTP turnover assays, or by biochemical data showing that the binding of purified Gi to Y1R is agonist dependent.

We appreciate the reviewer's suggestion. We performed GTP turnover assays using the purified $G\alpha_{i1}$ samples produced from E.coli and Sf9 cells, respectively. As shown here, $G\alpha_{i1}$ from E. coli is still functional. We have included this data in this revised manuscript (Supplementary Figure 29).

2. The binding of alpha5 of G α to Y1R seems a bit different from that in other G α -coupled GPCR structures. It may be helpful to compare the binding modes of G α in the structures with Y1R and other neuropeptide GPCRs including NTSR1 and mu-opioid receptor.

We thank the reviewer for the suggestion. In this revised manuscript, we have included a comparison of the binding modes of G α in complexes with Y1R, NTSR1, and mu-opioid receptor (Supplementary Fig. 7).

p7, *“Most of these interactions are conserved in the hNTSR1-G α and μ OR-G α structures (Supplementary Fig. 7). However, the relative position of the α 5 helix of G α bound to Y1R slightly differs by \sim 2 Å displacement of the C-terminus of G α , or \sim 8° tilt angle of the α 5 helix of G α compared to NTSR1-bound and μ OR-bound G α structures, respectively, when aligning the receptors (Supplementary Fig. 7).”*

3. The figure numbering needs to be revised. There is no Figure 1d.

We acknowledge the error. We corrected the errors on p7, p10, and p11.

4. Based on my understanding, all the MD simulation data shown in Supplementary Figures 12, 18, and 23 were from the same three repeated runs. It may be less confusing (at least to me) to put them in one figure as different panels.

Yes, all the MD simulations data shown in Supplementary Figures 12, 18, and 23 were from the same three repeated runs. We appreciate the reviewer's suggestion but combining three figures into one figure seems like too much data to fit into one figure. Furthermore, in this revised manuscript, we have added one more figure for the analysis of MD simulations data, providing a total of four pieces of information. So, putting all the data of MD simulations in one figure seems complicated, in our opinion. Therefore, as a compromise, in this revised manuscript, we have presented two separate figures of MD simulations data, one for analysis of MD simulations of NPY (supplementary figure 14) and the other for analysis of MD simulations of Y1R (supplementary figure 23).

5. The section "Difference in binding mode between antagonists and NPY C-terminal segment" may need more discussion of receptor activation mechanism, e.g. how the differences in the binding of antagonists and NPY lead to receptor activation. I would also suggest the authors revise the title of this section to indicate that there is a discussion of receptor activation mechanism, not just a description of differences in the binding of several different ligands.

We thank the reviewer's suggestion. Since Y1R is activated by the interaction with NPY C-terminus and NPY-N-terminus, we discuss the Y1R activation mechanism upon NPY binding at the discussion section after describing the interactions of both C-terminal and N-terminal regions of NPY. Nevertheless, we agree to some extent with the reviewer's comment that it would be good to discuss the structural changes of the TM core of Y1R and the activation mechanism upon binding of the NPY C-terminal tail in the section "Difference in binding mode between antagonists and NPY C-terminal segment". Therefore, as suggested by the reviewer, we revised the title of this section to "Structural changes in Y1R TM core by binding of NPY C-terminal tail" (p9) and discussed the major events of structural changes of Y1R TM during activation (p9-10) with new figure (Figure 4c).

p.9, ***“Structural changes in Y1R TM core by binding of NPY C-terminal tail”***

p.9-10, *“In both inactive structures, Q120^{3.32} makes van der Waals contact with M310^{7.43}, although Q120^{3.32} rotamers differ in two inactive structures, forming polar interaction with BMS-193885 or*

van der Waals interaction with W276^{6.48} (Fig. 4c). In the NPY-bound Y₁R structure, the Q120^{3.32} sidechain adopts an upward-facing rotamer, forming a polar contact with the amidated C-terminus. Also, Q120^{3.32} no longer interacts with M310^{7.43} (>7 Å) but with C93^{2.57}. Reorganization of interaction network near Q120^{3.32} by NPY binding stabilizes the conformation of the upward displacement of TM3 (Fig. 4c)."

p.10, *"In addition, the structural comparison of the TM core between inactive and active states of Y₁R suggests the key events of conformational changes during activation by NPY, rearrangement of the hydrophobic network around F286^{6.58} at the entrance to the ligand-binding pocket, and rearrangement of interaction network around Q120^{3.32} and I128^{8.40} in the connector region at the bottom of the ligand-binding pocket."*

6. The authors need to have a clearer definition of the N-terminal loop region, the helical region, and the C-terminal tail of NPY. It is quite confusing that the authors used these terms loosely in the discussion. For example, the authors stated that "the N-terminal region of Y₁R, residues 21–32, was crosslinked to NPY in a photo-crosslinking experiment", while 21-32 is the helical region. When the authors talked about the N-terminal region, I am not sure if they referred to the N-terminal loop region (Y1-P5), or the loop region with the un-modeled region, or the entire region Y1-I31 including the helical region. I also recommend including clear terms for the three different regions of NPY in Figure 1C.

We appreciate the reviewer's suggestion. We clarified the three regions of NPY as the N-terminal loop region (Y1-R19), the helical region (Y20-I31), and the C-terminal tail (T32-Y36), which are now shown in Figure 1C.

7. Is there any data showing that the extreme N-terminal region of NPY (Y1-P5) is important for the action of NPY? Why does it fold back to extend towards the receptor instead of sticking away from the receptor like other neuropeptides?

Although we did not use the N-terminal 5 residue truncation of NPY(NPY(6-36)) for signaling assay, we showed that the extreme N-terminal region of NPY (Y1-P2) is important for downstream signaling through Y₁R (Fig. 5b). Furthermore, we performed an additional signaling assay using mutant NPY, in which Y1 and P2 are replaced with A1 and A2 (denoted as NPY(AA-36)), demonstrating that tyrosine and proline at positions 1 and 2, respectively, are important for NPY signaling (Fig. 5b).

Interestingly, the extreme N-terminal region of NPY folds back and interacts with the receptor. We think the favorable interactions of Y1-P2 of NPY with the receptor, as shown in our cryo-EM structure, make this possible. However, for a more accurate interpretation, extensive biophysical studies, such as measuring enthalpy and entropy changes upon NPY binding, would be required, which is beyond the scope of this study.

Reviewer #2 (Remarks to the Author):

*** Summary of the research and overall impression ***

Park and co-workers report the first experimental structure of the Y1 receptor (Y1R) in an active conformation bound to the natural agonist peptide neuropeptide Y (NPY) and heterotrimeric Gi determined by cryo-EM. The structure shows that the C-terminal portion of NPY adopts an extended conformation and binds deep within the core of Y1R. The N-terminal regions of the peptide are less well resolved, but cryo-EM combined with mutagenesis and MD simulations suggests dynamic interactions of this region with the extracellular areas of the receptor. The structure rationalizes mutagenesis results presented in this manuscript as well as previously published mutagenesis and NMR studies and reveals the peptide-receptor interactions important for receptor activation.

The manuscript is very interesting and adds important insight into how natural peptide agonists activate their cognate GPCRs. The research is of very high quality and the manuscript is logically structured. However, the clarity of the manuscript should be enhanced by editing for language and writing quality to improve flow and readability. In addition, I suggest the authors address the following issues before publication.

Specific areas for improvement

*** Major issues ***

1. On p. 6, lines 1-5, the authors compare the solvent-accessible volumes of inactive-state and active-state Y1R. The relatively modest conformational changes in the extracellular ends of a subset of the TM helices described in the text and shown in Fig. 2a do not appear to explain the large increase in volume stated in the text (414 Å³ vs 859 Å³). The authors should verify and show more clearly where this 2-fold increase in volume originates from. Have unmodeled residues/sidechains been taken into account? Also, the volumes given in the main text do not match those given in Supplementary Fig. 5. Finally, the methods section on the calculation of the binding site volumes (p. 21) lacks information on what software was used.

As the reviewer pointed out, in our original manuscript, the solvent-accessible volume of NPY-bound Y1R was written as 791 Å³ in Supplementary Fig. 5, but in the text, it was written as 859 Å³. Moreover, we found errors in the previous calculation of the solvent-accessible volumes of the inactive and active states of Y1R. We apologize for these errors.

We recalculate the solvent-accessible volumes in the two inactive and NPY-bound active structures to 498 Å³, 515 Å³, and 730 Å³, respectively, showing a 1.4-fold increase in volume in the NPY-bound structure. This increase may be attributed to the C-terminal tail of NPY, which is bulkier than antagonists. We observed the outward movement of the extracellular tips of TM3, TM4, TM6, and TM7 by 1.9~2.5 Å in the NPY-bound Y1R, as described in the text (p6). In this new Supplementary Figure 5, we demonstrate the solvent-accessible volumes in top views and side views for clarity. We correct the calculated volumes in the main text of this revised manuscript as follows.

p6, *"In fact, the calculated solvent-accessible ligand-binding cavities in the antagonist-bound and NPY-bound structures are ~505 and 730 Å³, respectively (Supplementary Fig. 5)."*

No unmodeled sidechains were pointing toward the TM core. For example, in the Y1R structure, sidechains of residues 39, 41, 59, and 66 from TM1 are unmodeled, pointing outward. Similarly, in the antagonist-bound structures, unmodeled residues did not affect the volume calculation, as they are in the loop regions or at the N-terminal tip of TM1.

In the methods section, we have provided link to software code and fill out the code and software submission checklist.

p23, *"The python code for calculating the solvent accessible volume of the ligand-binding pocket is*

available at https://github.com/seoklab/GPCR_binding_cavity_volume_calculation.”

2. The authors should expand their discussion of selectivity. The molecular basis of selectivity of the three related peptide agonists NPY, PYY, and PP is briefly discussed on p. 13. According to the work by Cabrele et al. cited in the manuscript (ref. 37), PP does not activate Y1R at all or only with very low potency, but instead is selective for Y4R and to a lesser degree Y5R. Conversely, NPY is a potent agonist of all four neuropeptide Y receptors. Based on the structure of NPY-bound Y1R presented by the authors and amino-acid differences in the binding sites of Y1R and Y4R, is it possible to explain the observed selectivity and in particular the high potency of PP on Y4R, perhaps with the help of homology modeling?

We thank the reviewer’s suggestion. First, we performed signaling assay of Y1R using PP, showing that Y1R has a 64-fold increased EC₅₀ value for PP compared to NPY (Supplementary Fig. 25).

Then, we carefully examined amino acid differences in the ligand-binding sites of Y1R and Y4R by sequence alignment (Supplementary Fig. 27). Most NPY-interacting residues in Y1R are conserved in Y4R, but E288 replaces F286 in Y1R in Y4R. Of note, we propose that F286 of Y1R contributes to selectivity between Y1R and Y2R since V291 replaces F286 in Y2R. Homology modeling of Y4R suggests that E288 would form an extensive charged interaction network with R33 and R36 of NPY (Supplementary Fig. 28), providing sufficient interaction energy for Y4R to bind PP as well as NPY. This is just our speculation and should be validated by experimental data. We include this information in the text as follows.

p.14-15, *“Indeed, our signaling assay shows that Y₁R has a 100-fold increased EC₅₀ value for PP compared to NPY (Supplementary Fig. 25). Among four subtypes of the NPY receptor, Y₄R was activated in response to PP. Interestingly, Y₄R has Glu at the position of 6.58 (E288^{6.58}) instead of Phe as in Y₁R (Supplementary Fig. 27), suggesting that Y₄R would form a polar interaction network with nearby polar residues (E203^{4.52}, R211^{5.35}, T215^{5.39}, N285^{6.55}, E288^{6.58}, D289^{6.59}) and basic residues of NPY, R33, and R35 (Supplementary Fig. 28). We speculate that this extensive polar interaction network would provide sufficient interaction energy for Y₄R to accommodate PP as well as NPY, which should be validated with experimental data.”*

3. The proposed activation mechanism is an important finding of this work, but is currently only described in the legend of Fig. 6. The mechanism should be summarized in the discussion section of the main text.

We appreciate the reviewer’s suggestion. As suggested by the reviewer, we describe the proposed activation mechanism in the discussion section.

p.15-16, *“Comparison of the inactive and active structures suggests the activation mechanism of Y₁R upon NPY binding. --- At the bottom of the ligand-binding pocket, Y36 of NPY forms hydrophobic interaction with I124^{3.36} through the phenyl group and polar interaction with Q120^{3.32} through the amidated C-terminus, leading to a rotamer change of Q120^{3.32}. This is followed by a rotamer change of I128^{3.40}, which interacts with I124^{3.36}, and repacking of the side chains of P223^{5.50}, F272^{6.44}, and W276^{6.48}. A series of these changes upon NPY binding pulls TM3 upward and causes outward movement of the cytoplasmic region of TM6 (Fig. 6).”*

*** Minor issues ***

1. The information presented in Supplementary Fig. 26 could be enhanced by coloring the peptide

sequences by conservation.

We appreciate the reviewer's suggestion. In Supplementary Figure 26 of the original manuscript, conserved residues are highlighted in blue. (This figure is now renumbered as Supplementary Figure 25)

2. p. 6, line 7-10: "...extracellular loops ... and the N-terminal region ... reorient themselves extensively...". This statement should be supported with a figure.

We appreciate the reviewer's suggestion. Supplementary Figure 19 of the original manuscript shows the density of extracellular loops and the N-terminal region of Y1R surrounding the N-terminal and helical regions of NPY. We moved this figure to supplementary figure 6 to support the sentence on page 6, lines 7-10. In this sentence, the statement of "... reorient themselves extensively..." was deleted and moved to p13, line 6, and supported with Supplementary Figure 23. Supplementary Figure 23 shows that the N-terminal region and ECL2 form dynamic interactions with the helical region of NPY during MD simulations.

p.6, *"we could observe the density of the N-terminal and helical regions of NPY, which are surrounded by extracellular loops (ECLs) and the N-terminal region of Y1R (Supplementary Fig. 6)."*

p.13, *"In addition, ECL2 was demonstrated to interact with the helical region of NPY during the simulations, suggesting that ECL2 and the N-terminal region of Y1R form dynamic interactions with NPY by reorienting themselves extensively to accommodate the NPY binding (Supplementary Fig. 23)."*

3. p. 7, last line: "A similar acidic ligand-binding pocket is observed in OX2R,...". This is difficult to discern from Supplementary Fig. 11b. Given that the C-terminal segment of OxB is mostly composed of hydrophobic residues rather than basic sidechains as in NPY, this may also not be relevant. Please comment.

We regret the confusion our statement has caused. We tried to compare the surface charge of the receptor region where the C-terminus of each peptide is located. This is not related to the peptide sequence, but our message was not delivered properly by referring to R33 and R35 of NPY in the text. So, to clarify our message, we modified the sentences as below.

p.8, *"The acidic residues of the binding pocket repel the negatively charged carboxyl terminus and favor the amidated C-terminus. A similar acidic patch in the ligand-binding pocket is observed in OX2R, where the amidated C-terminus of OxB binds."*

4. p. 8, line 9: "...the phenyl ring of Y36 forms an intramolecular interaction with R35 of NPY,...". This is not obvious in Fig. 3a the authors refer to. The authors may want to consider adding another figure.

We thank the reviewer for the suggestion. We include another figure showing an intramolecular interaction between R35 and Y36 of NPY (Supplementary Figure 14).

5. p. 13, lines 9 and 10: The sentence "This study presents...based on our cryo-EM map" is unclear and needs rephrasing.

Thank you for pointing this out. We modified the sentence as follows.

p.13, *"This study presents a model candidate containing the five N-terminal residues of NPY, constructed based on our cryo-EM map."*

*** Other points ***

1. The adjectives C-terminal and N-terminal are used instead of the nouns C-terminus and N-terminus throughout the manuscript. Also, the authors should settle on the use of carboxyl terminus or C-terminus (amino-terminus or N-terminus).

Thank you for pointing this out. We corrected the errors and used “N-terminus” and “C-terminus” consistently throughout the manuscript.

2. p. 2, line 6: The sentence “The extended conformation...” does not make sense as it stands. It is not the conformation that binds in the receptor core, but the peptide.

Thank you for pointing this out. The sentence was modified as follows.

p.2, *“The NPY C-terminal segment forming the extended conformation binds deep into the Y₁R transmembrane core, -- “*

3. On several occasions, the authors refer to “Fig. 1d”, e.g. on p. 7, line 6. No panel d is included in Fig. 1. Should this read “Fig. 1c”?

We acknowledge the error. We corrected the errors on p7, p10, and p11.

4. p.5, line 13: “scFv16” should be introduced properly and a reference included.

Thank you for pointing this out. We introduce “scFv16” and add a reference on p5.

p.5, *“-- and single-chain variable fragment termed scFv16, that specifically recognizes heterotrimeric G_i was added as a stabilizer (reference 19)“*

5. p. 5, line 15: “GDN” should be defined.

Thank you for pointing this out. We include the full name of “GDN” on p5.

p.5, *“--- at a nominal resolution of 3.2 Å in glyco-diosgenin (GDN) micelles”*

Reviewer #3 (Remarks to the Author):

The paper by Choi and coworkers describes their cryo-electron microscopy study of the complex between the peptide NPY, the Y1 receptor, and Gi protein, together with ScFv16. The study is conducted at a high scientific level.

The authors point to the very significant potential of NPY, PYY3-36, and other peptides in this family, as potential therapeutics for the treatment of obesity, cancer and other diseases.

I am not a structural biologist but assessed the paper from a medicinal chemistry point of view. While some of their findings are in agreement with past reports on NPY-Y1R interactions, their study also provides new and detailed insights on how the peptide NPY binds the Y1 receptor. This will be very interesting for peptide medicinal chemists, both when working with NPY and when working with peptides from other, related families.

There are small mistakes in references 30 and 53.

Thank you for pointing this out. We corrected the error.

REVIEWERS' COMMENTS

Reviewer #1 (Remarks to the Author):

I appreciate the additional data provided by the authors. All of my concerns are appropriately addressed.

Reviewer #2 (Remarks to the Author):

The manuscript has improved, and this reviewer's comments and suggestions have been addressed satisfactorily.

Minor comments

1. The PDB identifier for NTS–NTSR1 given in Supplementary Fig. 24 is incorrect.
2. For consistency, the two instances of 'carboxyl terminus' should be changed to 'C-terminus' on p. 8 lines 160 and 163.
3. The link to the github repository containing the authors' code for the calculation of the volume of the GPCR binding pocket appears broken or not available yet.

Reviewer #3 (Remarks to the Author):

I appreciate the revisions and recommend that the paper is accepted.

We thank the reviewers for their comments.

Our point-by-point responses are written in blue below.

REVIEWERS' COMMENTS

Reviewer #1 (Remarks to the Author):

I appreciate the additional data provided by the authors. All of my concerns are appropriately addressed.

We thank the reviewer's comment.

Reviewer #2 (Remarks to the Author):

The manuscript has improved, and this reviewer's comments and suggestions have been addressed satisfactorily.

We thank the reviewer's comment.

Minor comments

1. The PDB identifier for NTS–NTSR1 given in Supplementary Fig. 24 is incorrect.

We thank the reviewer for pointing out this error. We corrected the error (PDB ID 7L1U  7L0Q).

2. For consistency, the two instances of 'carboxyl terminus' should be changed to 'C-terminus' on p. 8 lines 160 and 163.

We thank the reviewer for pointing this out. We changed 'carboxyl terminus' to 'C-terminus' on page 8 (lines 160 and 163) as suggested.

3. The link to the github repository containing the authors' code for the calculation of the volume of the GPCR binding pocket appears broken or not available yet.

Since our unpublished structural model is included in the reference file in the code (for reproducibility), we did not open the link yet. The link will be freely available, as soon as our paper is accepted.

Reviewer #3 (Remarks to the Author):

I appreciate the revisions and recommend that the paper is accepted.

We thank the reviewer's comment.